# VARIATIONAL MODE DECOMPOSITION AND LINEAR EMBEDDINGS ARE WHAT YOU NEED FOR TIME-SERIES FORECASTING

## ABSTRACT

Time-series forecasting often faces challenges due to data volatility, which can lead to inaccurate predictions. Variational Mode Decomposition (VMD) has emerged as a promising technique to mitigate volatility by decomposing data into distinct modes, enhancing forecast accuracy. This study integrates VMD with linear models to develop a robust forecasting framework. Our approach is evaluated on 13 diverse datasets, including ETTm2, WindTurbine, M4, and 10 air quality datasets from Southeast Asian cities. The effectiveness of the VMD strategy is assessed by comparing Root Mean Squared Error (RMSE) values from models utilizing VMD against those without it. Additionally, we benchmark linear-based models against well-known neural network architectures such as LSTM, BLSTM, and RNN. The results demonstrate a significant reduction in RMSE across nearly all models following VMD application. Notably, the Linear + VMD model achieved the lowest average RMSE in univariate forecasting at 0.619. In multivariate forecasting, the DLinear + VMD model consistently outperformed others, attaining the lowest RMSE across all datasets with an average of 0.019. These findings underscore the effectiveness of combining VMD with linear models for superior time-series forecasting.

## 1 INTRODUCTION

Time-series forecasting plays a critical role in predicting future events, especially when augmented by machine learning techniques. These models can be trained to achieve high predictive accuracy and low error rates by effectively leveraging large datasets. However, obtaining optimal results requires a rigorous preprocessing pipeline, including data cleaning, decomposition, and hyperparameter tuning. Moreover, the selection of the appropriate forecasting model is essential to balance computational efficiency with prediction performance.

One of the most widely used architectures in time-series forecasting is the Transformer, originally known for its strong performance in natural language processing (NLP), speech recognition, and computer vision tasks. Nevertheless, Zeng et al. (2022) have questioned the efficacy of Transformer models for time-series forecasting, sparking comparative performance evaluations against simpler linear-based models, such as Linear, NLinear, and DLinear. Empirical results indicate that these linear models frequently outperform Transformer-based approaches—including FEDFormer, Auto-Former, Informer, LogTrans, and PyraFormer—on key metrics like Mean Squared Error (MSE) and Mean Absolute Error (MAE) (Zeng et al., 2022). Linear models will be more effective if we are working with stationary or stable dataset.

Data decomposition techniques also significantly influence forecasting outcomes. Variational Mode Decomposition (VMD) is one such technique that has demonstrated effectiveness in reducing data volatility (Dragomiretskiy & Zosso, 2014). VMD mainly utilizes Lagrange multiplier, quadratic penalty, Hilbert transformation, and L2 Norm to minimize the bandwidth in data modes while keeping those modes compact with its center frequencies. A 2021 study comparing Recurrent Neural Network (RNN)-based models under various scenarios with PM2.5 air quality data from multiple cities in China highlights VMD's superior performance. The study evaluated models without decomposition, with Empirical Mode Decomposition (EMD), and using VMD. Results showed that VMD consistently achieved the lowest Root Mean Squared Error (RMSE), particularly when paired with Bidirectional LSTM models (Zhang et al., 2021). Another study in 2023 has implemented VMD together with LSTM model to predict Dissolved Oxygen (DO) and Total Phosphorus (TP) values for

water quality prediction. VMD implementation leads to reduced MAE, RMSE, and MAPE values. DO prediction result reduced by 38.8%, 41.4% and 44.4%, TP prediction result reduced by 30.4%, 34.4% and 32.2% respectively (Tao et al., 2023).

Given VMD's ability to mitigate data volatility and the tendency of linear models toward stable data, there is potential for more effective time-series forecasting, even under unstable data conditions. This work aims to extend the investigation into the comparative performance of these models against AutoRegressive Integrated Moving Average (ARIMA) and deep learning-based models such as RNN, Long Short-Term Memory (LSTM), and Bidirectional LSTM (BiLSTM), which remain prevalent in time-series forecasting research despite their increased complexity.

Motivated by these observations, this work investigates the integration of Variational Mode Decomposition with Linear Embeddings to enhance the robustness of time-series forecasting. The primary goal is to evaluate the effectiveness of VMD for data decomposition and assess the performance of simplistic linear models (Linear, NLinear, and DLinear) alongside RNN-based models in achieving minimal error rates, particularly under volatile data conditions.

## 2 RELATED WORKS

### 2.1 TIME-SERIES FORECASTING MODELS

**Statistical Methods** Time series data can be forecasted by simple statistical methods that doesn't require iterative learning. One of the most used techniques is AutoRegressive Integrated Moving Average (ARIMA), which can outperform complex structural models in time series forecasting. ARIMA can handle strong sequential correlations by utilizing its differential method (Liu et al., 2016). Seasonal ARIMA (SARIMA) incorporates seasonal autoregressive, differential method, and moving average to capture seasonal patterns within data. Other statistical methods besides ARIMA such as Exponential Smoothing (ETS) and Vector AutoRegression (VAR) are also commonly used in time series forecasting cases. Those methods can capture trends, seasonality, and time-dependent variables (Erkekoglu et al., 2020).

**Linear Models.** Linear models have their roots in regression analysis, where they are used to predict outcomes based on a linear relationship between independent and dependent variables. Their broader application in machine learning and pattern recognition was discussed by Bishop (2006), highlighting their simplicity and interpretability. However, traditional linear models often struggle to capture complex, non-linear patterns in data. Recent advancements, such as those by Zeng et al. (2022), introduced improved versions like NLinear and DLinear, which have proven competitive with Transformer-based models in time-series forecasting. Another notable linear model is RLinear, introduced in 2023 (Li et al., 2023), which employs Reversible Instance Normalization (RevIN) and the Channel Independent (CI) strategy to enhance performance. RevIN normalizes data instances independently, improving convergence rates and reducing overfitting (Kim et al., 2021), while CI normalizes each feature independently in multivariate time-series forecasting, leading to better feature-wise representation (Han et al., 2024).

**Neural Networks.** Neural networks are more complex algorithms, inspired by the structure of the human brain, consisting of layers of interconnected nodes (neurons) that model non-linear relationships. Trained using backpropagation, these models adjust weights to improve predictive performance (Aggarwal, 2018). While simple feedforward networks suffice for basic tasks, more complex architectures like Recurrent Neural Networks (RNNs) are used for sequential data. The main idea of RNN is the connection of recurrent hidden units back to themselves (Benidis et al., 2022). However, due to the limitations of RNNs in handling long-term dependencies, Long Short-Term Memory (LSTM) networks were developed as an improved version of RNN, offering an enhanced ability to retain and model long-term patterns in time-series data (Sherstinsky, 2020).

**Transformers.** The Transformer architecture represents a significant advancement in time-series forecasting, using a self-attention mechanism to weigh the importance of different parts of an input sequence relative to others. Unlike RNNs, Transformers process all input elements simultaneously, improving both efficiency and performance, especially in capturing long-range dependencies (Vaswani et al., 2023). Numerous transformer-based models, such as AutoFormer and Reformer, have been developed for time-series forecasting. AutoFormer decomposes data into trend and seasonal components and applies auto-correlation (Wu et al., 2021), while Reformer reduces computational complexity through local attention and reversible layers (Kitaev et al., 2020). Further

advancements include Stationary Transformer, which recovers intrinsic non-stationary information and unifies input statistics (Liu et al., 2022), and TimeSNet, which transforms 1D temporal variations into a 2D space, allowing the temporal evolution of one feature to influence others (Wu et al., 2023). Recently, PatchTST was introduced, leveraging Channel Independence (CI) and segmenting time-series inputs into subseries-level patches for improved representation (Nie et al., 2023).

**Large Language Models (LLMs).** Large Language Models (LLMs) are designed primarily to understand, generate, and work with natural language. However, recent advances have expanded their application to time-series forecasting and computer vision tasks by leveraging their robustness in pattern recognition and reasoning (Mirchandani et al., 2023). Popular models like GPT and BERT have been fine-tuned for time-series data, enabling them to perform tasks such as classification, anomaly detection, and forecasting through unified frameworks (Zhou et al., 2023). Time-LLM, introduced in 2023, further extends this capability by reprogramming time-series inputs into text-based representations, guiding LLMs with declarative prompts. Time-LLM has shown competitive performance against state-of-the-art models like GPT4TS, DLinear, and PatchTST (Jin et al., 2024).

## 2.2 DATA MODE DECOMPOSITION

Data mode decomposition is a critical technique for breaking down complex time-series data into multiple modes or Intrinsic Mode Functions (IMFs) with reduced complexity. One widely used method for data decomposition is Empirical Mode Decomposition (EMD), which adaptively decomposes a signal into a series of IMFs and a residual trend, capturing components with varying frequency ranges. EMD iteratively sifts through the data to identify and extract IMFs, making it particularly effective for non-stationary and nonlinear time series (Huang et al., 1998). However, EMD is known to be sensitive to noise and prone to mode mixing, a phenomenon where distinct modes are incorrectly merged.

Variational Mode Decomposition (VMD) was introduced as a more robust alternative to overcome these limitations. VMD formulates the decomposition task as a variational problem, aiming to minimize the bandwidth of each mode while ensuring the modes remain as distinct as possible. This process is achieved by utilizing the Lagrange multiplier and the Alternating Direction Method of Multipliers (ADMM) (Dragomiretskiy & Zosso, 2014). The Lagrange multiplier enforces constraints on the decomposed components through a penalty term $\lambda$, while ADMM simplifies the optimization by breaking down the Lagrange function into sub-problems, which are iteratively solved until convergence (Boyd et al., 2011a). Unlike EMD, VMD is more resistant to noise and reduces mode mixing by separating modes in the frequency domain through a structured optimization process.

Another prominent method for data decomposition is Dynamic Mode Decomposition (DMD), which focuses on extracting essential dynamic characteristics of the data. DMD achieves this by applying matrix analysis techniques and singular value decomposition (SVD) to represent the data in terms of its dynamic modes (Proctor et al., 2016). This approach allows DMD to capture the underlying dynamics of time-series data and is particularly effective in systems governed by linear or near-linear processes.

## 3 MODEL

### 3.1 VMD-LINEAR FRAMEWORK

The structure of VMD-Linear shown in Figure 1 was inspired by VMD-BLSTM framework introduced by Zhang et al. (2021). The goal is to reduce data bandwidth in all IMF or data modes with the sum of all IMF equal to that of the original data. LTSF-Linear models will be assigned on each IMF for time-series forecasting. As the final step, the individual forecasts will be aggregated to produce the overall forecast result, considering that the sum of all modes must be approximately equal to the original data.

### 3.2 VARIATIONAL MODE DECOMPOSITION

Variational Mode Decomposition (VMD) is a method commonly used to decompose complex signals into several k parts of signals (modes) that have been modulated in terms of amplitude and frequency, also known as Intrinsic Mode Function (IMF).

$$IMF \text{ or } u_k(t) = A_k(t)\cos(\phi_k(t)) \tag{1}$$

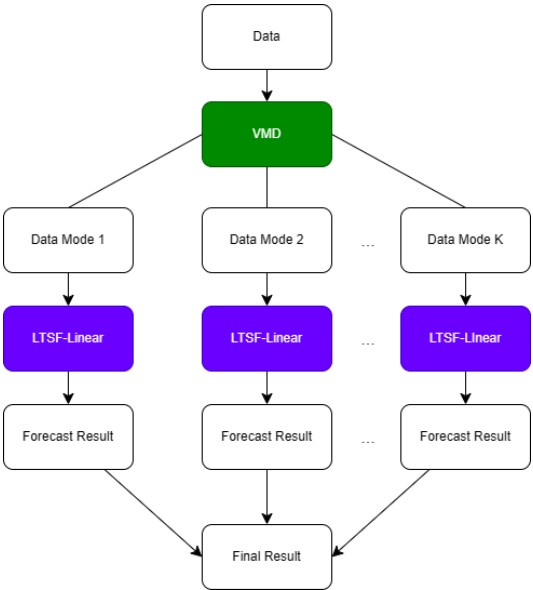

Figure 1: VMD-Linear Framework

The symbol $\phi_k(t)$ is a non-decreasing function (can be considered a wave) with the condition that $\phi_k(t) \geq 0$, and $A_k(t)$ is the envelope of the wave $\phi_k(t)$. All IMF formed must satisfy Equation 1. VMD applies Hilbert transform to obtain the analytic frequency spectrum $(Au_k)$ of each mode $u_k$. In the transformation process, each mode is multiplied by the center frequency $(e^{-jw_k t})$ to shift its frequency spectrum towards the baseband (Dragomiretskiy & Zosso, 2014).

$$Au_k(t) = [\delta(t) + \frac{j}{\pi t}] * u_k(t)e^{-jw_k t} \tag{2}$$

The Dirac distribution is symbolized as $\delta$ and j is the imaginary number satisfying $j^2 = -1$. The symbol $w_k$ in the center frequency $(e^{-jw_k t})$ denotes the central pulse/vibration of each mode. The signal's bandwidth will be estimated using $H^1$ Gaussian Smoothness of the demodulated signal. In this case, the estimation is performed by squaring the $L^2$ normalization of the gradient.

$$min_{u_k, w_k}\{\Sigma_k \parallel \partial_t[(\delta(t) + \frac{j}{\pi t}) * u_k(t)]e^{-jw_k t} \parallel_2^2\} \tag{3}$$

Equation 3 only applies when $\Sigma_k\ u_k = original\ signal$. The main objective of VMD is to minimize the bandwidth around the center frequency in each mode $u_k$ while maintaining the condition that the sum of all modes equals the original signal. The bandwidth of each mode needs to be compact around its center frequencies $(\omega)$. In order to achive that goal, the use of Lagrange multiplier $(\lambda)$ and quadratic penalty are recommended. The quadratic penalty is used to address additional Gaussian noise. In noise-free conditions, infinite weights are required to enforce strict data accuracy. The Lagrange multiplier is a way to impose strict constraints. Combining the Lagrange multiplier's strict constraint imposition and the quadratic penalty $(\alpha)$ with finite weights, the enhanced Lagrangian function L is formulated in Equation 4 to transform the constrained problem. Finding the most suitable $\alpha$ is a challenge when using VMD. If the alpha is set too high, there is a risk of failing to capture the center frequencies, while if it is set too low, it may result in excessive noise around the center frequencies (Dragomiretskiy & Zosso, 2014).

$$L(u_k, w_k, \lambda) = \alpha\Sigma_k \parallel \partial_t[(\delta(t) + \frac{j}{\pi t}) * u_k(t)]e^{-jw_k t} \parallel_2^2 +$$
$$\parallel f(t) - \Sigma_k u_k(t) \parallel_2^2 + [\lambda(t), f(t) - \Sigma_k u_k(t)] \tag{4}$$

The minimization problem in Equation 4 can be addressed using the Lagrange function L by employing the Alternate Direction Method of Multipliers (ADMM) iteratively on $u_k$, $w_k$, and $\lambda$. ADMM works by decomposing large-scale problem into smaller subproblems will be solved iteratively. Iteration will stop if the condition is met: $\Sigma_k(\| u_k^{n+1} - u_k^n \|_2^2 \ / \ \| u_k^n \|_2^2) < \varepsilon$ where $\varepsilon$ is the threshold value (Boyd et al., 2011b). Equation 5, 6, and 7 are subproblems decomposed by ADMM algorithm from Equation 4.

$$\widehat{u}_k^{n+1} = \frac{\hat{f}(w) - \Sigma_{i \neq k} \widehat{u}_i(w) + \frac{\widehat{\lambda}(w)}{2}}{1 + 2\alpha(w - w_k)^2} \tag{5}$$

$$\widehat{w}_k^{n+1} = \frac{\int_0^\infty w|\widehat{u}_k(w)|^2 dw}{\int_0^\infty |\widehat{u}_k(w)|^2 dw} \tag{6}$$

$$\hat{\lambda}^{n+1}(w) = \hat{\lambda}^n(w) + \tau(\hat{f}(w) - \Sigma_k \widehat{u}_k^{n+1}) \tag{7}$$

For multivariate forecasting, the main idea still identical but most variables is represented in vectors. Thus, increasing the complexity and computational demands (Rehman & Aftab, 2019).

### 3.3 Time Series Forecasting with Classic Neural Networks

Time Series Forecasting were made even more possible with the appearance of neural networks, especially RNN-based neural networks. RNN-based neural networks were designed to work with sequential data which was perfect for time series forecasting. However, those models are considered complex depending on the number of hidden units it has. With its complexity, only a computer with big enough computing resource can run those models.

**RNN Model.** The RNN (Recurrent Neural Network) model introduces a way to capture temporal dependencies in time series data. RNN updates its hidden state based on its current input ($X_t$), previous hidden state ($S_{t-1}$), and its activation function ($f$) which was shown in Equation 8 (Benidis et al., 2022). $W_x$ and $U_S$ are respectively weights for input and hidden state.

$$S_t = f(W_x.X_t + U_S S_{t-1} + b) \tag{8}$$

Its hidden state allows RNN to access historical information in order to make future predictions more accurate. The problem that lies with RNN model is the vanishing gradient problem which diminishes gradients exponentially as layers propagated each time.

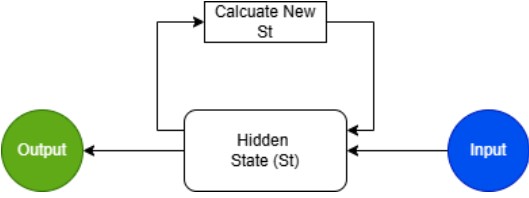

Figure 2: RNN Structure

**LSTM Model.** The LSTM (Long Short-Term Memory) model still has similar idea from RNN but it utilizes cell state and gates for handling long-term sequential data. As shown in Figure 3, LSTM has 3 gates which are input gate ($i_t$), forget gate ($f_t$), and output gate ($o_t$). By utilizing those gates for updating the cell state ($C_t$), LSTM can carry relevant information over long time-steps (Mahto et al., 2021).

$$i_t = \sigma(W_i.X_t + U_i C_{t-1} + b_i) \tag{9}$$

$$f_t = \sigma(W_f.X_t + U_f C_{t-1} + b_f) \tag{10}$$

$$o_t = \sigma(W_o.X_t + U_oC_{t-1} + b_o) \tag{11}$$

$$C_t = f_t \circ C_{t-1} + i_t \circ tanh(W_CX_t + U_CC_{t-1} + b_C) \tag{12}$$

$$Output = o_t \circ tanh(C_t) \tag{13}$$

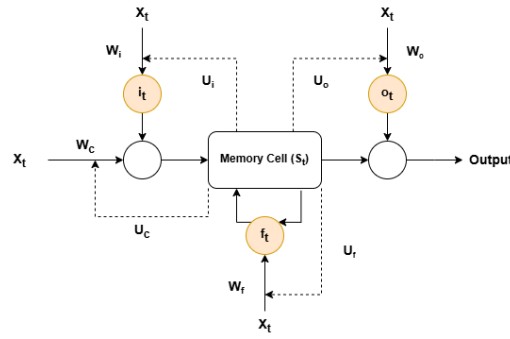

Figure 3: LSTM Structure

**BiLSTM Model.** BiLSTM (Bidirectional LSTM) is simply a LSTM structure that use 2 layers of LSTM, one layer going forward and the other going backward. For each time-step ($t$), BiLSTM can access both past and future information, providing a more complete understanding of the sequential data. Then, the output on both layers can be concatenated or transformed with other methods. Figure 4 illustrates the structure of BiLSTM.

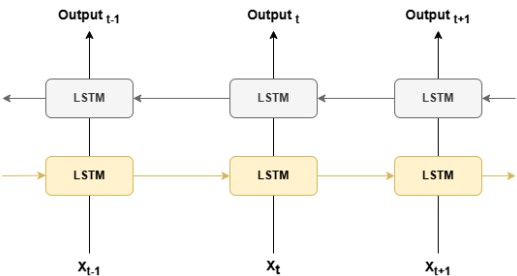

Figure 4: Bidirectional LSTM Structure

### 3.4 TIME SERIES FORECASTING WITH STATISTICAL METHOD

Time series forecasting can also be achieved with simple yet effective statistical methods that doesn't require hidden nodes or iterative learning. Its simplicity and effectiveness made this method used by many people, even until now.

**ARIMA.** AutoRegressive Integrated Moving Average (ARIMA) is one of those, it utilizes AutoRegressive to capture relationship between an observation and previous observations. Real world time series data are mostly non-stationary, transforming the data into stationary form can be achieved by utilizing 1st order differential method (shown in Equation 14), 2nd order differential method (shown in Equation 15), etc (Liu et al., 2016). The degree of differential method is denoted as $d$, whereas $X_t$ indicates observation data at timestep $t$.

$$\nabla X_t = X_t - X_{t-1} \tag{14}$$

$$\nabla^2 X_t = \nabla X_t - \nabla X_{t-1} \tag{15}$$

By integrating both AutoRegression (AR) and Moving Average (MA), the main equation of ARIMA is shown in Equation 16. $\beta$, $\alpha$, and $\epsilon$ are respectively AutoRegressive weights, Moving Average weights, and residual.

$$\nabla^d X_t = \Sigma_{i=1}^q \beta_i \epsilon_{t-i} - \Sigma_{i=1}^k \alpha_i \nabla^d X_{t-i} + \epsilon_t \tag{16}$$

### 3.5 TIME SERIES FORECASTING WITH LINEAR MODELS

Long-term Time Series Forecasting with Linear models (LTSF-Linear) refers to a family of lightweight linear models that include Linear, NLinear, and DLinear, designed for forecasting tasks over extended temporal horizons (Zeng et al., 2022). These models offer simplicity and computational efficiency while remaining competitive with more complex architectures such as Transformer-based models. Their design facilitates both univariate and multivariate forecasting by dynamically adjusting the number of channels in the parameter space, making them highly scalable and adaptable for different time-series tasks.

The key advantage of LTSF-Linear models lies in their simplicity. Unlike deep learning models, which often require extensive memory and computational resources, LTSF-Linear models typically employ no more than two linear layers. This reduction in model complexity not only accelerates training and inference times but also reduces the risk of overfitting, especially in datasets with limited or noisy observations. Furthermore, linear models are inherently interpretable, offering insights into the contribution of individual input features to the forecasted outcome, which is particularly valuable for time-series applications in areas like finance, healthcare, and climate science.

**NLinear Model.** As depicted in Figure 5, the NLinear (Normalized Linear) model is designed to handle distributional shifts that may occur in long-term forecasting. The model mitigates this issue by normalizing the residuals of the last input before passing it through the linear layer, which effectively reduces the gap between the training and test distributions. This step helps maintain forecast stability over time and improves robustness against concept drift, a common issue in real-world time-series data. After the transformation, the reduced input is added back before the final prediction is generated, ensuring that the model benefits from both the normalization and the retained original signal information. NLinear has been shown to outperform traditional linear models when applied to datasets with frequent shifts in data distributions (Zeng et al., 2022).

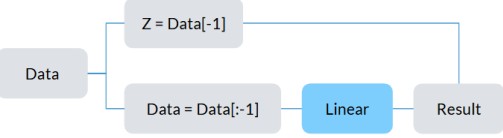

Figure 5: NLinear Framework

**DLinear Model.** The DLinear (Decomposition Linear) model, shown in Figure 6, introduces a decomposition step to address long-term dependencies more effectively. DLinear breaks the input time-series into two components: Trend and Seasonal. These components are processed separately through distinct linear layers. The trend component captures the gradual, long-term patterns in the data, while the seasonal component isolates repetitive, shorter-term fluctuations. By separating these two components, DLinear can model each behavior more precisely, yielding better forecasting results in datasets that exhibit clear seasonality or trends. This architecture is particularly beneficial for time-series data in domains such as energy demand forecasting, where trends and seasonal effects are often prominent (Zeng et al., 2022).

## 4 EXPERIMENTS

In this section, we present experiments using ARIMA, LTSF-Linear models, and classic neural network models such as Recurrent Neural Network (RNN), Long Short Term Memory (LSTM), and Bidirectional LSTM (BLSTM) on 13 real-world datasets. In this research, we compare each model performance using RMSE and MAE values.

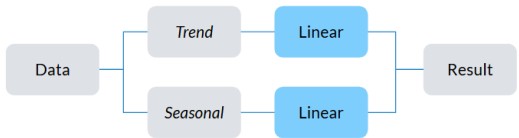

Figure 6: DLinear Framework

## 4.1 DATASET

We used 13 different real-world datasets obtained from various sources in the internet. The datasets are described as below:

1. **ETTm2 Dataset**, this dataset was compiled by Beijing Guowang Fuda Science and Technology Department (Zhou et al., 2021). It consists of electricity distribution dynamics and was captured in 15-minute intervals with a total of 69,680 data points.

2. **Wind Turbine**, The following data contains information about a certain wind turbine in Turkey that was captured using the SCADA system (Erisen, 2019). Wind Turbine-related data such as wind speed, wind direction, and power generated were captured in 10-minute intervals with a total of 50,500 data points.

3. **M4 Dataset**, The M4 dataset was originally used in M4 Competition as a benchmark to evaluate various models in real-world scenarios. It consists of many features and can be used for multivariate forecasting (Makridakis et al., 2020). This dataset was recorded in 7-day intervals.

4. **SEA Air Quality Dataset** We also used various real-world Air Quality Index (AQI) datasets from many cities recorded from 27 January 2022 to 22 Mei 2023 (AQI, 2023). Those cities were Central Singapore, Banjarbaru, Batam, Jakarta, South Jakarta, Jambi, Malang, Medan, Samarinda, and Semarang.

ETTm2, Wind Turbine, and M4 datasets were used because of its relatively high variance in their features (Arthur et al., 2024). Those 3 datasets will be used in multivariate forecasting together with the Central Singapore AQI dataset. As for univariate forecasting, all datasets will be used including mentioned multivariate datasets.

We divide the dataset into training data and test data with a certain ratio. A dataset with fewer than 500 rows will be split with a 90:10 ratio, and a dataset with more than 10000 rows will be trimmed to the first 10,000 rows and then split with an 80:20 ratio. This data trimming is necessary to reduce computational load, considering that the number of prediction models follows the number of $K$ modes. We conducted two different testing scenarios based on several features, namely univariate and multivariate. In the univariate testing scenario, we only used one feature on each dataset in this testing scenario. Meanwhile, the multivariate testing scenario used 4 datasets that contain multiple features such as ETTm2, Wind Turbine SCADA, M4 Competition, and Central Singapore Air Quality dataset.

## 4.2 EXPERIMENTAL PROTOCOLS

First, datasets were cleaned and then normalized by using Standard Scaler. After normalization, variational mode decomposition was implemented on the dataset and will create K data modes or IMFs. Each data mode will go through a separate forecasting model, and then the forecasting results of each mode is summed together to attain the final forecasting result. We used LASSO regularization in loss calculation to avoid vanishing gradients that could happen while training the models (Mairal & Yu, 2012). RMSE was used as a metric evaluation method for emphasizing each model's impact on reducing error rates. We also used MAE for its robustness against outliers (Hodson, 2022).

## 4.3 FORECASTING RESULTS

**Baseline** we choose other well-known neural network models such as Recurrent Neural Network (RNN), Long Short Term Memory (LSTM), and Bidirectional LSTM (BiLSTM). RNN is able to recognize patterns in sequences of data, but cannot handle long-term dependencies. RNN were

### Table 1: Multivariate RMSE Results

| Model | M4 Weekly (413 Data) | Central Singapore (481 Data) | Wind Turbine (10000 Data) | ETTm2 (10000 Data) | Average RMSE |
|---|---|---|---|---|---|
| Linear | 1.041 | 1.203 | 1.397 | 1.380 | 1.255 |
| VMD + Linear | 0.102 | 0.098 | 0.007 | 0.010 | 0.054 |
| DLinear | 1.063 | 1.221 | 1.404 | 1.406 | 1.273 |
| VMD + DLinear | **0.040** | **0.024** | **0.005** | **0.007** | **0.019** |
| NLinear | 1.116 | 1.331 | 1.414 | 1.398 | 1.314 |
| VMD + NLinear | 0.064 | 0.051 | 0.011 | 0.014 | 0.035 |
| RNN | 1.092 | 1.204 | 1.381 | 1.321 | 1.249 |
| VMD + RNN | 0.080 | 0.077 | 0.330 | 0.243 | 0.182 |
| LSTM | 1.070 | 1.165 | 1.379 | 1.355 | 1.242 |
| VMD + LSTM | 0.092 | 0.124 | 0.025 | 0.252 | 0.123 |
| BiLSTM | 1.126 | 1.218 | 1.394 | 1.379 | 1.279 |
| VMD + BiLSTM | 0.089 | 0.203 | 0.038 | 0.031 | 0.090 |
| ARIMA | 1.009 | 1.078 | 0.975 | 1.036 | 1.024 |
| VMD + ARIMA | 4.786 | 4.472 | 4.137 | 6.280 | 4.918 |

### Table 2: Multivariate MAE Results

| Model | M4 Weekly (413 Data) | Central Singapore (481 Data) | Wind Turbine (10000 Data) | ETTm2 (10000 Data) | Average MAE |
|---|---|---|---|---|---|
| Linear | 0.407 | 1.090 | 1.057 | 1.100 | 0.913 |
| VMD + Linear | 0.086 | 0.095 | 0.004 | 0.008 | 0.048 |
| DLinear | 0.398 | 0.986 | 1.066 | 1.088 | 0.884 |
| VMD + DLinear | **0.026** | **0.022** | **0.003** | **0.005** | **0.007** |
| NLinear | 0.459 | 1.277 | 1.081 | 1.094 | 0.977 |
| VMD + NLinear | 0.065 | 0.040 | 0.009 | 0.011 | 0.015 |
| RNN | 0.447 | 0.920 | 1.060 | 1.037 | 0.866 |
| VMD + RNN | 0.058 | 0.058 | 0.015 | 0.189 | 0.08 |
| LSTM | 0.424 | 0.851 | 1.090 | 1.005 | 0.842 |
| VMD + LSTM | 0.068 | 0.112 | 0.021 | 0.199 | 0.1 |
| BiLSTM | 0.480 | 1.073 | 1.095 | 1.091 | 0.934 |
| VMD + BiLSTM | 0.077 | 0.146 | 0.025 | 0.023 | 0.067 |
| ARIMA | 0.642 | 0.846 | 0.804 | 0.821 | 0.778 |
| VMD + ARIMA | 2.697 | 3.490 | 3.394 | 4.694 | 3.568 |

further upgraded to LSTM as an algorithm that can recognize patterns in long sequences of data while maintaining useful information (Sherstinsky, 2020). To enable LSTM to use both future and past contexts of the data, Bidirectional LSTM were developed (Graves et al., 2005). We also used ARIMA as a baseline model by considering its simplicity and effectiveness in time series forecasting.

**Results** Multivariate forecasting results were listed in Table 1 and 2. RMSE and MAE values reduction in multivariate predictions were shown after applying VMD with most models, except ARIMA. This situation occurs because there is a possibility that the model we use is unable to capture the patterns or trends in the data, this requires further hyperparameter tuning. The lowest RMSE values were obtained by the DLinear + VMD model in all prediction cases, with an average RMSE and MAE of 0.019 and 0.007. Without VMD, LTSF-Linear models tends to perform close to RNN-based

### Table 3: Univariate RMSE Results

| Model | Banjarbaru PM2.5 (481 Data) | Batam PM2.5 (481 Data) | Jakarta PM2.5 (481 Data) | South Jakarta PM2.5 (481 Data) | Jambi PM2.5 (481 Data) | Malang PM2.5 (481 Data) | Medan PM2.5 (481 Data) | Samarinda PM2.5 (481 Data) | Semarang PM2.5 (481 Data) | M4 Weekly V2 (413 Data) | Central Singapore PM10 (481 Data) | Wind Turbine ActivePower (10000 Data) | ETTm2 HUFL (10000 Data) | Average RMSE |
|---|---|---|---|---|---|---|---|---|---|---|---|---|---|---|
| Linear | 1.030 | 1.038 | 1.054 | 0.998 | 0.988 | 0.985 | 1.036 | 0.851 | 1.023 | 0.594 | 1.039 | 1.012 | 1.015 | 0.974 |
| VMD + Linear | 0.620 | 0.605 | **0.563** | **0.662** | 0.529 | **0.581** | 0.545 | **0.510** | 0.824 | 0.631 | **0.574** | **0.786** | 0.788 | **0.619** |
| DLinear | 1.031 | 0.996 | 1.034 | 1.050 | 0.994 | 1.018 | 1.008 | 0.819 | 1.003 | 0.602 | 1.077 | 1.005 | 1.023 | 0.973 |
| VMD + DLinear | 0.703 | 0.687 | 0.779 | 0.841 | 0.521 | 0.855 | 0.714 | 0.652 | 0.780 | 0.753 | 0.605 | 1.022 | 0.908 | 0.742 |
| NLinear | 1.225 | 1.412 | 1.209 | 1.225 | 1.284 | 1.157 | 1.161 | 0.951 | 1.115 | 0.654 | 1.498 | 1.411 | 1.365 | 1.205 |
| VMD + NLinear | 0.908 | 0.858 | 0.713 | 0.922 | 0.712 | 0.953 | 0.962 | 0.634 | 1.122 | 0.613 | 0.906 | 1.295 | 0.794 | 0.883 |
| RNN | 1.171 | 0.960 | 1.335 | 1.294 | 1.186 | 1.268 | 1.145 | 0.937 | 1.215 | 0.651 | 1.481 | 1.404 | 1.387 | 1.187 |
| VMD + RNN | 0.494 | 0.894 | 0.694 | 1.141 | 1.101 | 1.161 | 0.440 | 0.796 | **0.631** | 0.698 | 0.922 | 0.919 | 1.004 | 0.824 |
| LSTM | 1.208 | 1.289 | 1.365 | 1.130 | 1.191 | 1.133 | 1.141 | 0.908 | 1.169 | 0.616 | 1.213 | 1.454 | 1.397 | 1.170 |
| VMD + LSTM | **0.445** | **0.398** | 1.274 | 0.814 | 1.108 | 1.222 | 0.572 | 0.844 | 0.901 | 0.307 | 0.642 | 1.940 | **0.744** | 0.872 |
| BiLSTM | 1.239 | 1.273 | 1.262 | 1.195 | 1.082 | 1.173 | 1.191 | 0.884 | 1.286 | 0.608 | 1.347 | 1.416 | 1.414 | 1.182 |
| VMD + BiLSTM | 0.726 | 0.866 | 0.827 | 1.122 | 1.011 | 1.260 | **0.415** | 1.435 | 0.959 | **0.151** | 0.938 | 0.865 | 1.129 | 0.881 |
| ARIMA | 1.012 | 0.924 | 1.092 | 0.994 | 1.002 | 0.990 | 0.985 | 0.972 | 1.019 | 1.003 | 1.174 | 0.964 | 1.011 | 1.010 |
| VMD + ARIMA | 1.119 | 1.036 | 1.028 | 0.942 | **0.491** | 0.767 | 0.865 | 0.758 | 1.063 | 0.601 | 0.972 | 0.968 | 0.923 | 0.887 |

Table 4: Univariate MAE Results

| Model | Banjarbaru PM2.5 (481 Data) | Batam PM2.5 (481 Data) | Jakarta PM2.5 (481 Data) | South Jakarta PM2.5 (481 Data) | Jambi PM2.5 (481 Data) | Malang PM2.5 (481 Data) | Medan PM2.5 (481 Data) | Samarinda PM2.5 (481 Data) | Semarang PM2.5 (481 Data) | M4 Weekly V2 (413 Data) | Central Singapore PM10 (481 Data) | Wind Turbine ActivePower (10000 Data) | ETTm2 HUFL (10000 Data) | Average MAE |
|---|---|---|---|---|---|---|---|---|---|---|---|---|---|---|
| Linear | 0.820 | 0.826 | 0.829 | 0.764 | 0.819 | 0.808 | 0.848 | 0.682 | 0.861 | 0.423 | 0.796 | 0.937 | 0.776 | 0.783 |
| VMD + Linear | 0.494 | **0.463** | 0.545 | **0.568** | **0.380** | 0.447 | **0.383** | **0.485** | 0.654 | 0.406 | **0.407** | 0.878 | 0.729 | **0.526** |
| DLinear | 0.831 | 0.778 | 0.820 | 0.806 | 0.826 | 0.845 | 0.813 | 0.666 | 0.870 | 0.422 | 0.868 | 0.937 | 0.785 | 0.789 |
| VMD + DLinear | 0.460 | 0.492 | 0.543 | 0.591 | 0.447 | 0.563 | 0.517 | 0.576 | **0.571** | 0.479 | 0.490 | **0.748** | 0.709 | 0.552 |
| NLinear | 0.998 | 1.088 | 0.944 | 0.952 | 1.063 | 0.911 | 0.956 | 0.754 | 0.886 | 0.445 | 1.168 | 1.113 | 1.065 | 0.949 |
| VMD + NLinear | 0.565 | 0.732 | 0.674 | 0.687 | 0.575 | 0.795 | 0.698 | 0.695 | 0.865 | 0.569 | 0.623 | 0.891 | 0.823 | 0.707 |
| RNN | 0.893 | 0.783 | 1.091 | 1.049 | 0.975 | 1.021 | 0.971 | 0.715 | 1.002 | 0.423 | 1.182 | 1.084 | 1.085 | 0.944 |
| VMD + RNN | 0.722 | 0.545 | 0.602 | 0.997 | 1.205 | 1.425 | 0.584 | 0.591 | 0.613 | 0.587 | 0.550 | 1.111 | 1.127 | 0.819 |
| LSTM | 0.945 | 0.941 | 1.043 | 0.903 | 0.985 | 0.943 | 0.943 | 0.708 | 0.943 | 0.468 | 0.927 | 1.146 | 1.093 | 0.922 |
| VMD + LSTM | **0.266** | 0.995 | 0.956 | 0.592 | 0.993 | **0.374** | 0.851 | 0.915 | 0.573 | 0.619 | 0.541 | 0.903 | 1.148 | 0.748 |
| BiLSTM | 1.000 | 0.992 | 1.070 | 0.918 | 0.901 | 0.916 | 0.933 | 0.705 | 1.086 | **0.391** | 1.101 | 1.104 | 1.112 | 0.940 |
| VMD + BiLSTM | 0.632 | 0.921 | **0.446** | 0.774 | 1.747 | 0.438 | 0.571 | 0.639 | 0.880 | 0.892 | 0.790 | 0.862 | **0.561** | 0.781 |
| ARIMA | 0.809 | 0.732 | 0.911 | 0.785 | 0.815 | 0.816 | 0.819 | 0.769 | 0.875 | 0.664 | 0.889 | 0.882 | 0.775 | 0.810 |
| VMD + ARIMA | 0.991 | 0.855 | 0.852 | 0.822 | 0.411 | 0.647 | 0.680 | 0.610 | 0.920 | 0.454 | 0.722 | 0.898 | 0.697 | 0.735 |

models but worse. VMD played a big role in reducing volatility in data and minimizing bandwidth in data.

Univariate results were shown in Table 3 and 3. In most datasets, the use of VMD decomposition methods has proven to significantly enhance the performance of prediction models. However, there are still few cases where the model did not experience a decrease in RMSE, such as in the prediction of WindTurbine ActivePower data with LSTM, which saw an increase in RMSE from 1.421 to 1.940. However, MAE values for the same case has shown the opposite, MAE values decrease from 1.146 to 0.903. There is a possibility that outliers within data modes caused RMSE values to increase, since RMSE is sensitive to outliers. These outliers appear from the results of VMD decomposition, considering that VMD focuses only on reducing data instability and keeping the sum of all modes similar from the original data, and not on reducing outliers. To minimize outliers within data modes, further hyperparameter tuning for $K$ and $\alpha$ is necessary.

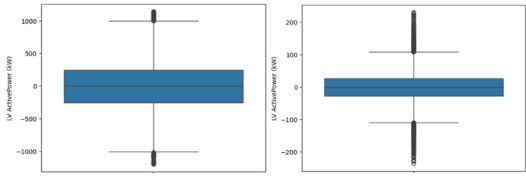

Figure 7: Outliers in Few Data Modes

The best univariate prediction model based on average RMSE across all prediction cases was VMD + Linear model with an average RMSE and MAE of 0.619 and 0.526, followed by VMD + DLinear model in second place with an average RMSE and MAE of 0.742 and 0.552 respectively.

## 5    CONCLUSION AND FUTURE WORK

This study demonstrates the effectiveness of Variational Mode Decomposition (VMD) in improving time-series forecasting by decomposing complex signals into multiple data modes. By integrating VMD with the LTSF-Linear models—Linear, DLinear, and NLinear—we leveraged the simplicity and competitive performance of these models compared to ARIMA and classic Neural Network approaches. Our evaluation, using RMSE and MAE as a metric evaluation method, compared VMD-enhanced linear models with ARIMA and established neural networks, including RNN, LSTM, and BLSTM. The results clearly show that VMD significantly reduces RMSE in both univariate and multivariate forecasting tasks. Specifically, the Linear + VMD model achieved the best performance across all univariate datasets, while the DLinear + VMD model excelled in multivariate tasks. These

findings underline the robustness and superior performance of VMD-enhanced linear models, presenting a compelling avenue for future research in time-series forecasting.

Future work should focus on refining the LTSF-Linear models for long-term time-series forecasting and also implementing VMD on other widely used models such as FEDformer, OneNet, etc.

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
