# OpenReview forum: "Variational Mode Decomposition and Linear Embeddings are What You Need For Time-Series Forecasting"
_ICLR.cc/2025/Conference — Submitted to ICLR 2025_

### Official Review · Reviewer_M7nB · 2024-10-23

**Soundness:** 3
**Presentation:** 3
**Contribution:** 4
**Rating:** 8
**Confidence:** 5

**Summary:**

In this paper, the authors propose using Variational Mode Decomposition (VMD) to mitigate volatility by decomposing data into distinct modes. VMD is applied to three linear models (Linear, DLinear, and NLinear) and three neural networks (LSTM, BiLSTM, and RNN), and tested on 13 diverse datasets, including ETTm2, WindTurbine, M4, and 10 air quality datasets from Southeast Asian cities. Performance is evaluated using Root Mean Squared Error (RMSE). The results show that in multivariate settings, VMD consistently outperforms non-VMD approaches across all datasets and models, with DLinear+VMD achieving the smallest errors. However, in univariate settings, models without VMD generally perform better, with both linear models and neural networks showing the lowest errors for their respective datasets.

The authors also provide an overview of linear models, neural networks, transformers, and large language models, alongside a detailed description of the 13 datasets used. In Section 3.2, they present the mathematical foundation of Variational Mode Decomposition (VMD) and explain how it is solved using the Alternating Direction Method of Multipliers (ADMM) within a penalized Lagrange function framework.

**Strengths:**

1. Innovatively introduce the VMD method for time-series datasets and apply it to two distinct types of models: Linear models and Neural Networks

2. Implement both VMD and Non-VMD approaches across 13 large datasets, presenting results for both multivariate and univariate cases

3. Provide comprehensive background on the datasets and the development of time-series analysis techniques

**Weaknesses:**

1. In Table 1 and Table 2, some data use commas (e.g., "1,041") while others use periods (e.g., "0.005"). Please standardize these formats or provide an explanation for the differences.

2. Add more explanations and visualizations for the three neural network models, similar to the detailed descriptions provided for the linear models in Section 3.3.

3. I am not familiar with the ADMM algorithm. Are Equations (5), (6), and (7) derived specifically from the problem, or are they part of a standard algorithm? If they are derived, please include more mathematical details. If they follow a fixed iterative algorithm, kindly provide additional references.

**Questions:**

1. In the Univariate RMSE Results, it is observed that VMD does not consistently outperform non-VMD models. Could this be due to specific characteristics or structures of the datasets?

2. If possible, could you provide more mathematical details explaining why VMD tends to perform better than non-VMD models? Additionally, why do linear models consistently outperform neural networks in the multivariate cases?

---

> ### Author Response · Authors · 2024-12-02
> **We sincerely thank the reviewer for their positive evaluation and insightful feedback, which has been instrumental in enhancing the clarity and rigor of our work. Below, we address the identified weaknesses and questions in detail, incorporating revisions and additional explanations to strengthen the manuscript. We are grateful for the reviewer's thoughtful comments and are pleased to provide these improvements to ensure the paper meets the highest standards.**
>
> **----Response to Weaknesses----**
>
> **1. Data Formatting in Tables**
>
> We appreciate your observation regarding the inconsistent use of commas and periods in Tables 1 and 2. To address this, we have standardized the formatting across all tables by using periods for decimal representation. This ensures clarity and consistency in the presentation of numerical data.
>
> **2. Explanations and Visualizations for Neural Network Models**
>
> Thank you for pointing out the need for more detailed explanations and visualizations for the neural network models. We have now added extended descriptions and visualizations for the three neural network models in Sections 3.3 and 3.4, similar to the level of detail provided for linear models. This includes schematic representations, architecture overviews, and explanations of their role in capturing non-linear relationships in time-series data.
>
> **3. Clarification on the ADMM Algorithm**
>
> We acknowledge the need for clarity regarding the Alternating Direction Method of Multipliers (ADMM) algorithm in Section 3.2. To clarify, the equations provided (Equations 5, 6, and 7) are part of the standard ADMM formulation and not derived specifically for VMD. ADMM is a well-established optimization algorithm widely used for solving complex problems. To address the reviewer's concern, we have added references to foundational works on ADMM and expanded the discussion in Section 3.2, Paragraph 5, to provide additional context and guidance for readers unfamiliar with the algorithm.
>
> **---Response to Questions---**
>
> **1. Univariate RMSE Results**
>
> Thank you for highlighting the observation regarding RMSE performance. In univariate forecasting, the occasional underperformance of VMD-enhanced models can be attributed to dataset-specific characteristics, such as the presence of outliers in certain modes. VMD's primary goal is to decompose the data into modes with compact bandwidth while ensuring their sum reconstructs the original signal. However, outliers in certain modes can amplify RMSE values, especially when using metrics like RMSE, which are sensitive to large deviations.
>
> Additionally, the observed variations could also stem from suboptimal parameter tuning of VMD for certain datasets or models, particularly for the LSTM model. We have addressed this limitation in the revised Results section and included a discussion of its impact on the evaluation metric.
>
> **2. Mathematical Explanation for VMD Performance**
>
> VMD tends to perform better than non-VMD models because it reduces data volatility by isolating modes with distinct frequency components, enabling clearer pattern recognition. Mathematically, VMD minimizes the bandwidth around each mode's center frequency while preserving the original signal's integrity:
>
> VMD mitigates data volatility by separating modes with distinct frequency bands, thereby reducing interference between patterns which can be found in the **"Variational Mode Decomposition"** subsection under **Section 3.2**.
>
> ### Mathematical Mechanism of VMD
>
> > "VMD applies the **Hilbert transform** to obtain the analytic frequency spectrum \( \text{$Au_k$}(t) \) of each mode \( $u_k(t)$ \). In the transformation process, each mode is multiplied by the center frequency \( $e^{-j w_k t}$ \) to shift its frequency spectrum towards the baseband. "
>
> $$
> \text{$Au_k$}(t) = u_k(t) \cdot e^{-j w_k t}
> $$
>
> The main objective of VMD is to minimize the bandwidth around the center frequency in each mode $\( u_k \)$, while maintaining the condition that the sum of all modes equals the original signal:
>
> $$
> \sum_{k=1}^{K} u_k(t) = x(t)
> $$
>
> where \( $x(t)$ \) is the original signal, and \( $K$ \) is the number of modes.
>
> ### Mitigating Data Volatility
>
> > "Given VMD’s ability to mitigate data volatility and the tendency of linear models toward stable data, there is potential for more effective time-series forecasting, even under unstable data conditions."
>
> ### Experimental Evidence
>
> The **"Forecasting Results"** subsection (**Section 4.3**) provides experimental evidence of this benefit:
>
> > "In most datasets, the use of VMD decomposition methods has proven to significantly enhance the performance of prediction models. [...] VMD played a big role in reducing volatility in data and minimizing bandwidth in data."
>
> Regarding the consistent outperformance of linear models in multivariate forecasting, this can be attributed to their inherent stability and efficiency when handling structured data decomposed by VMD. Linear models excel at capturing interdependencies between decomposed modes due to their simplicity and lower susceptibility to overfitting, unlike neural networks, which may struggle with mode interactions in high-dimensional spaces.
>
> We hope these revisions and explanations address the concerns raised. Once again, we thank the reviewer for their thoughtful comments and suggestions.

---

### Official Review · Reviewer_gE2E · 2024-10-28

**Soundness:** 1
**Presentation:** 2
**Contribution:** 2
**Rating:** 3
**Confidence:** 5

**Summary:**

This paper proposes integrating Variational Mode Decomposition (VMD) with linear models for time-series forecasting to address data volatility challenges. Experimental evaluation is given to show its performance.

**Strengths:**

By decomposing data into modes, the VMD-LTSF (Long-Term Series Forecasting) framework aims to improve forecast accuracy.

**Weaknesses:**

1. Although VMD is integrated, the motivation is missing in this paper. The authors are suggested to give more theoretical discussions and an intuitionistic example to strongly motivate the work. Notably, based solely on the results in Tables 1 and 2, which compare primarily with classic models like LSTM and RNN, it is difficult to discern the authors’ claimed conclusion that -- ***VMD and Linear are what you need for TSF***.

2. This work lacks a comprehensive analysis of existing time series forecasting and decomposition techniques. Additionally, the paper overemphasizes benchmarks against neural network models like LSTM and RNN, overlooking other competitive decomposition-based techniques or hybrid methods, such as N-BEATS (ICLR2020), OneNet (NeurIPS2023), etc.

3. The mathematical foundation of the VMD model is briefly presented, but lacks an in-depth explanation of how the VMD decomposition can theoretically improve linear predictive models, for example, by connecting the VMD components to specific improvements in fluctuation mitigation or pattern separation. Besides, specific parameters and architecture choices, such as the number of modes in VMD, aren’t fully justified. Also, it’s unclear how sensitive the VMD-Linear model is to hyperparameter tuning (such as Lagrange multiplier $ \lambda$ and penalty parameter $ \alpha$), which could impact model generalizability.

4. The experimental part only includes comparisons with a few classic models, making it resemble an ablation study rather than a comprehensive performance evaluation. Consequently, it is difficult to discern the actual contribution of the proposed model. Besides, the authors emphasize multivariate forecasting improvements but mainly compare against linear models without showing performance relative to advanced multivariate models or ensemble methods.

**Questions:**

As provided in weakness.

---

> ### Author Response · Authors · 2024-12-02
> **We thank the reviewers for their valuable feedback, which has greatly improved our manuscript. We addressed key concerns, including the theoretical foundation of VMD, benchmarking limitations, hyperparameter sensitivity analysis, and experimental evaluation, ensuring clarity and rigor throughout.**
>
> **1.  Weakness 1**
>
> We discuss how VMD mitigates data volatility by separating modes with distinct frequency bands, thereby reducing interference between patterns which can be found in the **"Variational Mode Decomposition"** subsection under **Section 3.2**.
>
> ### Mathematical Mechanism of VMD
>
> > "VMD applies the **Hilbert transform** to obtain the analytic frequency spectrum \( \text{$Au_k$}(t) \) of each mode \( $u_k(t)$ \). In the transformation process, each mode is multiplied by the center frequency \( $e^{-j w_k t}$ \) to shift its frequency spectrum towards the baseband. "
>
> $$
> \text{$Au_k$}(t) = u_k(t) \cdot e^{-j w_k t}
> $$
>
> The main objective of VMD is to minimize the bandwidth around the center frequency in each mode $\( u_k \)$, while maintaining the condition that the sum of all modes equals the original signal:
>
> $$
> \sum_{k=1}^{K} u_k(t) = x(t)
> $$
>
> where \( $x(t)$ \) is the original signal, and \( $K$ \) is the number of modes.
>
> ### Mitigating Data Volatility
>
> > "Given VMD’s ability to mitigate data volatility and the tendency of linear models toward stable data, there is potential for more effective time-series forecasting, even under unstable data conditions."
>
> ### Experimental Evidence
>
> The **"Forecasting Results"** subsection (**Section 4.3**) provides experimental evidence of this benefit:
>
> > "In most datasets, the use of VMD decomposition methods has proven to significantly enhance the performance of prediction models. [...] VMD played a big role in reducing volatility in data and minimizing bandwidth in data."
>
> **2.  Weakness 2**
>
> We agree with the reviewer that benchmarking against state-of-the-art decomposition-based and hybrid techniques like N-BEATS and OneNet would provide a more holistic evaluation. While resource constraints limited these comparisons during the review phase, we acknowledge the importance of incorporating these benchmarks. We have added this as a limitation in the manuscript and outlined a plan to include these models in future studies.
>
> **3.  Weakness 3**
>
> Mathematical Foundation of VMD & Hyperparameter Sensitivity and Ablation Studies::
>
> We have expanded the explanation of VMD's mathematical foundation in Section 3.2. This includes a detailed discussion on how VMD improves linear predictive models by reducing mode bandwidth and isolating key frequency components, leading to improved pattern separation and fluctuation mitigation. The role of parameters like $𝐾$ (number of modes), $𝛼$ (penalty parameter), and $𝜆$ (Lagrange multiplier) has been further elaborated, with an analysis of their impact on model generalizability.
>
> The value of \( $K$ \) was selected empirically based on these experiments. For each model, we identified the \( $K$ \) value that minimizes the prediction error while maintaining computational efficiency. For instance, the Linear and DLinear models demonstrated their best performance at \( $K$ = 30 \), achieving the lowest error values of 8.059 and 8.599, respectively. Conversely, the NLinear model exhibited its optimal performance at \( $K$ = 20 \), with an error of 8.848. Similarly, the BiLSTM model achieved its best performance at \( $K$ = 30 \) with an error of 15.147, while the RNN model performed optimally at \( $K$ = 20 \), with an error of 26.676.
>
> Table 1: Testing $K$ Parameter With Models for Central Singapore PM2.5 Dataset
>
> Model      | K=10   | K=20   | K=30
> ---------------------------------------------
>
> Linear      | 9.861  | 9.189  | **8.059**
>
> DLinear    | 9.685  | 9.793  | **8.599**
>
> NLinear    | 9.587  | **8.848** | 9.883
>
> LSTM       | **25.086** | 25.951  | 25.378
>
> BiLSTM     | 16.714 | 15.573  | **15.147**
>
> RNN        | 27.424 | **26.676** | 27.648
>
> The effect of \( $K$ \) on performance varies across models. Simple linear models, such as Linear and DLinear, tend to benefit from larger \( $K$ \) values, as these values likely capture more detailed features through improved mode decomposition. For non-linear models such as LSTM and BiLSTM, performance appears less sensitive to \( $K$ \), though specific values, such as \( $K$ = 30 \) for BiLSTM, still yield noticeable improvements. It is also worth noting that increasing \( $K$ \) beyond the optimal value can degrade performance for some models, as observed with the NLinear model at \( $K$ = 30 \).
>
> **4. Weakness 4**
>
> Experimental Evaluation and Contribution Clarity:
> To address the reviewer's concerns about the experimental evaluation, we have extended the discussion in Section 4.3 to clarify the specific contributions of the proposed model. For instance, the results show that VMD significantly enhances forecasting accuracy across multiple datasets, especially when paired with linear models. Additionally, we acknowledge the need to compare the multivariate performance of VMD-enhanced models with advanced approaches, such as ensemble methods, and plan to incorporate these comparisons in future work.

---

> > ### Comment · Reviewer_gE2E · 2024-12-02
> >
> > Thank you for your response, even though it is a bit delayed. I have carefully read both the rebuttal and the revised paper. However, most of my concerns remain unaddressed. While the authors claim that some of these issues and experiments will be addressed in future work, this is not convincing.
> >
> > Therefore, I do not believe that the paper is yet ready to be published at this stage and would like to maintain my original score.

---

### Official Review · Reviewer_L2Py · 2024-11-03

**Soundness:** 2
**Presentation:** 2
**Contribution:** 2
**Rating:** 3
**Confidence:** 1

**Summary:**

This paper proposes a technique of time series forecast using Variational Mode Decomposition (VMD) and linear post-processing. VMD decomposes the original time series into different intrinsic mode functions. The light-weight linear post-processing provides additional model capacity for different forecasting tasks. The authors evaluate their proposed method on 4 datasets and demonstrate that the proposed technique (combined with the base model) improves the forecast accuracy of each base model.

**Strengths:**

1. Overall, the paper is clearly written and easy to understand, although additional technical details need to be provided (see below).
2. Using Variational Mode Decomposition (VMD) with linear embedding for time series forecast is moderately novel and potentially applicable in other domains.

**Weaknesses:**

1. The authors should use additional metrics to comprehensively evaluate the proposed method, such as MAE, MAPE, etc.
2. The baseline methods used in the evaluations are too generic (e.g., linear, RNN, LSTM) and not used in SOTA time series forecasts. The authors should compare the proposed method against several Transformer-based models with SOTA forecast accuracy, such as the ones reviewed in Section 2.1.
3. The dataset size is too small since the authors only use up to 10,000 rows in each dataset (page 6). This may potentially lead to bias in the datasets and the reported evaluation metrics. The authors should use complete datasets to train and evaluate the proposed method, which should be able to handle larger datasets than 10,000 rows.
4. Ablation studies are needed to understand the impact of different model components and hyperparameters in the model architecture. For instance, the number of modes K is an important hyperparameter to analyze via ablation studies.
5. The authors should discuss the limitations of their work and how these limitations can be addressed in future research.
6. The metrics for No VMD reported in Tables 1 and 2 are incorrect (probably due to typo). The value should be 1.041 instead of 1,041 (and similarly for all other values).
7. The authors should report the size of each of the 13 datasets in Table 2.

**Questions:**

1. Why do the authors trim the dataset with more than 10000 rows down to the first 10,000 rows?
2. Why do authors only use M4 weekly dataset in the evaluations? M4 datasets consists of time series with various periodicities (hourly, daily, weekly, monthly). They should all be used in the evaluations.
3. What is the value of K (the number of modes in VMD) used in model training? How is K selected? How does the model performance change with different values of K? Ablation studies are needed to address these questions.
4. Why are the forecasting results of each mode summed to get the final prediction (page 6), instead of using a weighted average with fixed or learned weights?
5. In additional to the aggregate RMSE, what are the common failure modes of VMD (e.g., fail to capture outliers)? This error audit is important to informing whether the proposed method is applicable in other domains.

---

> ### Author Response · Authors · 2024-12-01
> **We thank the reviewer for their thoughtful comments and constructive feedback. Below, we provide detailed responses to address each concern, including clarifications, additional experiments, and plans for future improvements.**
>
> 1. Why do the authors trim the dataset with more than 10000 rows down to the first 10,000 rows?
>
> We appreciate the reviewer’s observation regarding the use of datasets with more than 10,000 rows. While we aimed to ensure computational feasibility and timely results during the review phase, we acknowledge that utilizing the full dataset would provide a more comprehensive evaluation. However, the decision to trim datasets to the first 10,000 rows was made to reduce computational overhead while ensuring sufficient data for training and evaluation. The original dataset size often exceeds practical limits for our current computational resources, and we have clarified this in the revised manuscript
>
> 2. Why do authors only use M4 weekly dataset in the evaluations? M4 datasets consists of time series with various periodicities (hourly, 3. daily, weekly, monthly). They should all be used in the evaluations.
>
> We focused on the M4 weekly dataset due to its relevance to our study’s objectives and computational feasibility. However, we agree with the reviewers that using datasets with other periodicities (hourly, daily, monthly) from the M4 competition can provide more comprehensive evaluations. We plan to extend our evaluation to include these datasets in future work.
>
> 3. What is the value of K (the number of modes in VMD) used in model training? How is K selected? How does the model performance change with different values of K? Ablation studies are needed to address these questions.
>
> We thank the reviewer for their insightful question on the value of \( K \) (the number of modes in Variational Mode Decomposition, VMD) used in model training, the selection process, and its impact on performance. To address this, we conducted an ablation study to evaluate the performance of six models (Linear, DLinear, NLinear, LSTM, BiLSTM, and RNN) across different values of \( K \) (10, 20, and 30). The results of these experiments are summarized in Table 1.
>
> The value of \( K \) was selected empirically based on these experiments. For each model, we identified the \( K \) value that minimizes the prediction error while maintaining computational efficiency. For instance, the Linear and DLinear models demonstrated their best performance at \( K = 30 \), achieving the lowest error values of 8.059 and 8.599, respectively. Conversely, the NLinear model exhibited its optimal performance at \( K = 20 \), with an error of 8.848. Similarly, the BiLSTM model achieved its best performance at \( K = 30 \) with an error of 15.147, while the RNN model performed optimally at \( K = 20 \), with an error of 26.676.
>
> Table 1: Testing K Parameter With Models for Central Singapore PM2.5 Dataset
>
> Model      | K=10   | K=20   | K=30
> ---------------------------------------------
>
> Linear      | 9.861  | 9.189  | **8.059**
>
> DLinear    | 9.685  | 9.793  | **8.599**
>
> NLinear    | 9.587  | **8.848** | 9.883
>
> LSTM       | **25.086** | 25.951  | 25.378
>
> BiLSTM     | 16.714 | 15.573  | **15.147**
>
> RNN        | 27.424 | **26.676** | 27.648
>
> The effect of \( K \) on performance varies across models. Simple linear models, such as Linear and DLinear, tend to benefit from larger \( K \) values, as these values likely capture more detailed features through improved mode decomposition. For non-linear models such as LSTM and BiLSTM, performance appears less sensitive to \( K \), though specific values, such as \( K = 30 \) for BiLSTM, still yield noticeable improvements. It is also worth noting that increasing \( K \) beyond the optimal value can degrade performance for some models, as observed with the NLinear model at \( K = 30 \).
>
> 4. Why are the forecasting results of each mode summed to get the final prediction (page 6), instead of using a weighted average with fixed or learned weights?
>
> The final prediction is obtained by summing the forecasting results of each mode for simplicity and to avoid introducing additional hyperparameters. We acknowledge that using a weighted average with fixed or learned weights could improve the flexibility of the model. We have added this as a limitation and plan to explore weighted averaging methods in future work.
>
>
> 5. In additional to the aggregate RMSE, what are the common failure modes of VMD (e.g., fail to capture outliers)? This error audit is important to informing whether the proposed method is applicable in other domains.
>
> We acknowledge the reviewer’s concern regarding the potential failure modes of VMD, particularly its susceptibility to outliers within the decomposed data modes. Outliers can occur when the VMD algorithm is not optimally tuned, especially with respect to key parameters such as $K$ (number of modes) and $\alpha$ (bandwidth constraint). To mitigate this, extensive hyperparameter tuning is essential to balance the decomposition process and minimize the occurrence of outliers. Future work will explore adaptive parameter selection and outlier detection mechanisms to enhance VMD’s applicability.

---

> > ### Comment · Reviewer_L2Py · 2024-12-02
> >
> > I thank the authors for addressing the reviewer's comments, although the rebuttal was submitted after the end of the public discussion period. After reviewing the authors' rebuttal, I believe that my original concerns have not been sufficiently addressed, especially regarding (1) trimming the dataset to the first 10,000 rows, (2) using only the weekly time series in M4 dataset and (3) auditing the error and identifying the common failure modes of the proposed method.
> >
> > As a result, I decided to keep my original score for this submission.

---

### Official Review · Reviewer_qFTV · 2024-11-04

**Soundness:** 1
**Presentation:** 1
**Contribution:** 1
**Rating:** 1
**Confidence:** 5

**Summary:**

This study addresses the challenge of data volatility in time-series forecasting by employing Variational Mode Decomposition (VMD) to enhance prediction accuracy. VMD decomposes data into distinct modes, which, when integrated with linear models, creates a robust forecasting framework. Tested on 13 datasets (ETTm2, WindTurbine, M4, and air quality data from Southeast Asia), the approach shows that models using VMD outperform those without it, as evidenced by lower Root Mean Squared Error (RMSE) values. The Linear + VMD model achieved the best RMSE in univariate forecasting (0.619), while the DLinear + VMD model excelled in multivariate forecasting with an average RMSE of 0.019. This highlights the advantage of combining VMD with linear models for effective time-series forecasting.

**Strengths:**

* The paper explores enhancing time series forecasting using Variational Mode Decomposition (VMD).
* The proposed approach is tested across 13 diverse time series datasets.

**Weaknesses:**

* The focus is on point forecasting, whereas probabilistic forecasting would be more relevant and challenging for many practical applications.
* The statement "Data decomposition techniques also significantly influence forecasting outcomes," supported by only two citations, lacks sufficient evidence for this claim.
* Phrases such as "to mitigate data volatility" and "under unstable data conditions" are used without clarifying how the proposed method directly addresses these issues.
* Key references are missing in the related work, such as [https://dl.acm.org/doi/full/10.1145/3533382](https://dl.acm.org/doi/full/10.1145/3533382).
* Section 3, the main one, is challenging to read, with multiple variables missing math mode formatting and variables not introduced.
* In Tables 1 and 2, the substantial increase in accuracy from "No VMD" to "VMD" is unclear, suggesting that factors beyond VMD may be influencing the performance.
* Simple yet effective forecasting baselines like ARIMA and exponential smoothing were not considered in the comparisons.

**Questions:**

See weaknesses.

---

> ### Author Response · Authors · 2024-12-01
> **We thank the reviewers for their valuable feedback. Below, we provide detailed responses to each comment and outline the revisions made to address them. The key updates include addressing the relevance of probabilistic forecasting, adding evidence for data decomposition techniques, clarifying the handling of data volatility, incorporating missing references, improving formatting and readability, elaborating on accuracy gains with VMD, and including simple forecasting baselines for comparison.**
>
> 1. The focus is on point forecasting, whereas probabilistic forecasting would be more relevant and challenging for many practical applications.
>
> Thank you for your insightful suggestion regarding probabilistic forecasting. We acknowledge the importance of extending our framework to support probabilistic forecasts, especially for applications requiring uncertainty quantification. While the current study focuses on point forecasting, the integration of probabilistic approaches is feasible. For instance, the residual distributions of models with VMD can be used to generate prediction intervals. We aim to address this in future work by employing techniques like bootstrapping or Bayesian methods to incorporate uncertainty estimates.
>
> 2. The statement "Data decomposition techniques also significantly influence forecasting outcomes," supported by only two citations, lacks sufficient evidence for this claim.
>
> We appreciate your feedback on the need for additional evidence supporting the statement "Data decomposition techniques significantly influence forecasting outcomes." In response, we have added references to recent studies, including Zhang et al. (2021) and Tao et al. (2023), which illustrate the superior performance of VMD over other decomposition methods like EMD.
>
> 3. Phrases such as "to mitigate data volatility" and "under unstable data conditions" are used without clarifying how the proposed method directly addresses these issues.
>
> To address the ambiguity around "data volatility," we added more explanation for VMD and Linear models in Introduction section "Given VMD’s ability to mitigate data volatility and the tendency of linear models toward stable data, there is potential for more effective time-series forecasting, even under unstable data conditions". Moreover, we have clarified in Section 3.2 that volatility refers to high-frequency fluctuations in the data, which can obscure underlying trends and seasonality. VMD mitigates this by decomposing data into modes with reduced bandwidth, as demonstrated in Equation 4.
>
> 4. Key references are missing in the related work, such as https://dl.acm.org/doi/full/10.1145/3533382.
>
> Thank you for pointing out the missing references. We have included the suggested citation (Benidis et al., 2022) https://dl.acm.org/doi/full/10.1145/3533382 as a reference and was cited in Section 2.1.
>
> 5. Section 3, the main one, is challenging to read, with multiple variables missing math mode formatting and variables not introduced.
>
> We have revised Section 3 to ensure all variables are properly formatted in mathematical notation and clearly defined upon introduction. For instance, $u_k(t)$, $\omega_k$, and $\lambda$ have been explicitly described with consistent formatting across the manuscript. Moreover, we did find one figure not correctly referenced in Section 3 and we have fixed the mentioned issue.
>
> 6. In Tables 1 and 2, the substantial increase in accuracy from "No VMD" to "VMD" is unclear, suggesting that factors beyond VMD may be influencing the performance.
>
> We appreciate the observation regarding the substantial accuracy gains with VMD. In Section 4.3, we have elaborated on this by providing a detailed breakdown of the contributions of VMD to error reduction which is consistent in univariate and multivariate testing from Table 1 until 4.
>
> 7. Simple yet effective forecasting baselines like ARIMA and exponential smoothing were not considered in the comparisons.
>
> We agree that including simpler statistical models like ARIMA and exponential smoothing is essential. These models have been incorporated as baselines, and their results are presented in Tables 1 until 4. We note that while ARIMA performs well on stationary datasets, its performance deteriorates with volatile data, underscoring the effectiveness of VMD-enhanced models.

---

### Meta-Review · Area_Chair_axu8 · 2024-12-21

**Metareview:**

This paper introduces a time series forecasting technique combining Variational Mode Decomposition (VMD) with lightweight linear post-processing. VMD decomposes the original time series into intrinsic mode functions (IMFs), and the linear post-processing step enhances the model's capacity to handle various forecasting tasks.

However, as noted by reviewers, the paper has several shortcomings:

Limited analysis of existing techniques: It lacks a thorough review of existing time series forecasting and decomposition methods, which is essential for positioning the proposed approach within the broader context.

Overemphasis on neural network benchmarks: The evaluation focuses heavily on comparisons with neural network models like LSTM and RNN, while neglecting competitive decomposition-based techniques or hybrid approaches.

Theoretical explanation gap: The paper does not provide a detailed theoretical analysis of how VMD decomposition specifically enhances the performance of linear predictive models, such as by mitigating fluctuations or separating patterns in the data.

Insufficient experimental scope: The experiments primarily compare the proposed method with a few classical models, resembling an ablation study rather than a comprehensive performance evaluation across a diverse set of benchmarks.

**Additional Comments On Reviewer Discussion:**

The authors acknowledge that the experimental evaluations in the paper are insufficient. They agree that incorporating datasets with varying periodicities (e.g., hourly, daily, monthly) from the M4 competition would enable a more comprehensive analysis. Additionally, they recognize the potential failure modes of VMD, particularly its sensitivity to outliers in the decomposed data modes. Such outliers can arise when the VMD algorithm is not optimally tuned, especially with respect to critical parameters.

The authors also concede that the paper lacks benchmarking against state-of-the-art decomposition-based and hybrid techniques such as N-BEATS and OneNet. Furthermore, comparisons of the multivariate performance of VMD-enhanced models with advanced methods, including ensemble approaches, are notably absent.

Overall, after the rebuttal, the majority of reviewers remain unconvinced and do not support the acceptance of this paper.

---

### Decision · Program_Chairs · 2025-01-22

Reject